# Does China’s Eco-Province Policy Effectively Reduce the Pollutant Emission Intensities?

**DOI:** 10.3390/ijerph191711025

**Published:** 2022-09-03

**Authors:** Mingfang Tang, Ling Li, Tao Li, Yuejing Rong, Hongbing Deng

**Affiliations:** 1State Key Laboratory of Urban and Regional Ecology, Research Center for Eco-Environmental Sciences, Chinese Academy of Sciences, Beijing 100085, China; 2University of Chinese Academy of Sciences, Beijing 100049, China; 3Northwest Institute of Eco-Environment and Resources, Chinese Academy of Sciences, Lanzhou 730000, China

**Keywords:** Eco-province, policy effect, pollutant emission intensity, time-varying difference-in-differences model, mediating effect model, China

## Abstract

Economic development and environmental conservation are two important challenges for China. A series of initiatives including Eco-province (EP) policies have been taken to achieve sustainable development by the Chinese government. Increasing concerns regarding policy implications on sustainable development have increased attention to the topic. However, the research on the relationship between economic development and pollutant (COD, SO_2_) emission intensities after the implementation of EP policy remains inadequate. We applied a Time-Varying Difference-in-Differences Model by employing Chinese provincial panel data to quantitatively study the policy effect, and further utilized the Mediating Effect Model to analyze the mechanism. The article generates several findings: (1) The EP policy had overall inhibitory effects on both COD and SO_2_ emission intensities, and it would reduce the emission intensity by 4.99% and 6.77% on average, respectively. However, there was a five year lag in the policy effect. (2) The policy effect was significant in the western and central provinces with high pollutant emission intensities, but not in the eastern provinces. (3) The primary inhibiting mediating effects of Eco-province policy to lower pollutant emission intensity were increased GDP per capita and inventions.

## 1. Introduction

Economic development and environmental conservation are two key challenges for China, and they are also urgent problems worldwide. China has made a succession of gains in reducing poverty and combating climate change, while playing an important international role. In 2015, China’s per capita GDP was still lower than the world average, less than one fifth of the average of developed countries, and more than 70 million people still lived below the national poverty line. By 25 February 2021, China had announced the attainment of comprehensive poverty alleviation. With the rapid economic expansion, China is facing unprecedented environmental pressure. To overcome the contradiction of economic development and environmental conservation, the Chinese government needs to achieve maximum economic development at the lowest environmental cost.

China has actively pushed the strategic planning and execution of sustainable development. In 1995, the Ministry of Ecology and Environment (MEE) published the *Outline of National Ecological Demonstration Zone Construction Plan (1996–2050)* [1]. The outline of the plan required not only the strong development of social economy, but also the rational development and usage of resources and the active protection of the natural environment. The Ecological Demonstration Zone contains three dimensions: Eco-county, Eco-city, and Eco-province. The construction contents and associated indicators are formulated, respectively [2]. Rather than focusing on a single ecological environment approach, the indicators take into account economic development, ecological environment protection, and social advancement. It adheres to the notion of locally independent and voluntary involvement and fully acknowledges the role of local governments. This province-level ecological civilization construction effort in China, which is unique from other local environmental regulations, piqued our interest in the potential effects of the program.

After being authorized as the first Eco-province (EP) pilot in 1999, Hainan province started looking into the creation of a provincial-level ecological pilot. Up until now, the state has successively launched 16 pilot EPs, including Hainan, Jilin, Heilongjiang, Fujian, Zhejiang, Shandong, Anhui, Jiangsu, Hebei, Liaoning, Guangxi, Tianjin, Sichuan, Shanxi, Henan, and Hubei. From the east, the middle, and then the west, the pilot EPs are gradually launched. Each province has released a construction planning blueprint for the EP, and according to that plan, the EP pilots will need 20 to 30 years to reach their objective by 2020 or 2030. Zhejiang province, which had been accepted as a pilot province in 2003, passed the pilot EP acceptance inspection in May 2020 and became China’s first EP.

EP is a regional policy implemented by each pilot province. The creation of EP policies and oversight of their execution fall under the purview of the province governments. In addition to legislation and official orders, policy also refers to the administrative tools employed to carry out policy objectives [3]. Previous studies have shown that complex environmental governance typically encompasses several actors and regions, with overlapping authority structures in various system components [4,5,6,7]. The quality of an environmental policy is dependent on a number of variables, including its own design, aims, tools, processes, centralized authority, and implementation. Therefore, the impact of environmental policy implementation is frequently complicated and fraught with scientific uncertainty. Institutionally, people in charge of preserving the environment and those in charge of running the economy are kept apart. Governments face significant difficulty in creating policies that produce greater results for less money [8].

There is disagreement in the current research on how environmental policy affects environmental quality. Some academics have unfavorable and critical opinions on the function and results of environmental regulation [9,10,11]. The implementation of any public policy needs to rely on policy-implementation agencies and implementers, including government departments, enterprises, community groups, individuals, etc. According to Austrian Economics, entrepreneurship is what drives economic phenomena [12,13,14]. However, state-interventionist policies could discourage innovative entrepreneurship and increase the cost of the industry. They urge the use of market-based environmental solutions. [15,16,17]. Public choice theorists argue, neither the state nor the market is uniformly successful in enabling individuals to sustain long-term, productive use of natural resource systems. Public policy sometimes would result in a zero-sum game. And communities of individuals are an alternative solution of governing the commons [18,19,20,21]. Psychological effects are also related to public policy. Entrepreneurs are more willing to invest and innovate when they feel confident. High political and economic uncertainty of public policy will make companies lose the will to innovate [22]. However, many academics stress that the government’s role is beneficial to long-term environmental protection and economic progress [23,24,25,26,27].

Since the 1960s, with the increasingly prominent problems of resource scarcity and environmental pollution, the relationship between economic growth and environmental quality has attracted extensive attention of the governments and scholars [23,28,29,30,31,32,33]. In the past two decades, the most representative study on the relationship between economic development and environmental pollution is the Environmental Kuznets Curve (EKC) [34]. A large number of scholars found an inverted-U shape relationship between pollutant emissions and economic development in both short and long term [35,36,37,38,39]. At the same time, through empirical studies, some scholars believed that the EKC hypothesis was difficult to be supported in the research area. [40,41,42,43]. Different research perspectives, data and econometric models may lead to different empirical conclusions on the EKC hypothesis. China has a very large land area, with great differences in regional economic and social development in the eastern, central, and western parts of the country. The variables ignored in the model can deviate the results making it difficult to obtain reliable conclusions. Therefore, in reflecting the relationship between economic development and environmental pollution, the EKC method maybe not the most appropriate to analyze the situation in China.

Several publications from the previous two decades have discussed the evaluation criteria and methodology for EP [44,45,46]. Some researchers have identified the policy effect from the dimension of a single province, such as Hainan [47], Heilongjiang [48], Zhejiang [49], and Fujian [50]. Some scholars analyze the policy effect of EP from the standpoint of the entire nation. Studies already conducted have identified geographical variations in EP construction and shown that eastern China has a higher level of ecological construction than central and western regions [51]. Comprehensive EP legislation can support ecological innovation and foster economic growth [52]. The policy has significantly reduced carbon emissions [53], and has showed the increase in ecological footprint per capita [54]. Furthermore, academics offer suggestions for improvement, believing that there are still some issues with the policy implementation process [55]. 

The emission intensity of COD and SO_2_ thus far has received far too little consideration. Pollutant emission intensity index is defined as pollutant emission per unit of GDP. This indicator, which is comparable across regions and time periods, directly shows the environmental cost paid for its economic growth. Due to the lower intensity of pollutant emissions, the region’s economic growth causes less environmental harm, which is more consistent with the objective of sustainable development [56]. Pollutant emission intensity is explicitly included as one of the nine binding metrics in the EP building indicators. The COD emission intensity should be less than 5.0 kg/10^4^ Chinese Yuan (CNY), and the SO_2_ emission intensity should be less than 6.0 kg/10^4^ CNY, per the index requirements. This study’s specific goal was to examine how province environmental policies in China affected the relationship between regional environment and economic development, regional heterogeneity, and the mechanism of such influence by introducing COD and SO_2_ emission intensity as the starting point.

Figure 1 shows the trend changes in China’s overall pollution emissions and emission intensities. As demonstrated, China’s overall COD emissions decreased between 1997 and 2017. The significant increase in 2011 is mainly because of the change of the statistical caliber of total COD discharge. The overall SO_2_ emission fluctuates greatly correlated with economic growth. From 1997 to 2017, the emission intensities of COD and SO_2_ exhibited a consistent lower trend, showing a drop in the environmental input per unit of economic output and a significant increase in China’s overall environmental efficiency. 

Figure 2 displays the distribution of pollutant emission intensities in pilot and non-pilot provinces in the years 2000, 2005, 2010, and 2015. Both in pilot EPs and non-pilot provinces, it demonstrates the declining trend in emission intensities. Pollutant emission intensity varies greatly between provinces, however. Particularly between 2000 and 2005, there was a large discrepancy between the provinces, however between 2010 and 2015, this difference significantly decreased. Pollutant emission levels across provinces show a trend toward convergence, however the rate of convergence varies [57].

(1) Is it possible for the EP pilot area to lower pollutant emission intensities more successfully than non-pilot provinces? (2) Does the policy impact on pollutant emission intensities vary by region? (3) What is the mechanism by which policy is implemented? This study offers fresh empirical proof of the structure of Chinese environmental governance. In the conflict between economic development and environmental protection, it can also serve as a model for other emerging nations.

The sections of this document are grouped using the following structure: The study techniques and data are mostly introduced in Section 2. The empirical findings are reported and discussed in Section 3. The research conclusion and policy suggestions are presented in Section 4.

## 2. Materials and Methods

### 2.1. Quasi-Experimental Analysis with Time-Varying DID

Sixteen pilot Eps were successively granted from 1999 to 2014, making up about half of all the provinces (excluding Hong Kong, Macao, and Taiwan). This study views the EP pilot project as a quasi-natural experiment of provincial ecological policy. To assess the efficacy of sustainable development, COD emission intensity and SO_2_ emission intensity are used as two typical indicators. To calculate the effectiveness of the policy, we use the quasi-experimental method of the Time-Varying difference-in-differences (Time-Varying DID) model.

The basic principle behind the DID model is to use a counterfactual framework to assess the changes in observed variables under the two scenarios of policy occurrence and non-occurrence. We split the 31 provinces into two groups: the treatment group (16 pilot EPs) that were subject to the policy, and the control group (15 non-pilot provinces). If the pilot start-up time is categorized as the same year, it is challenging to draw valid findings because the EPs’ pilot approval times vary, and the approval time period is rather long (from 1999 to 2014). As a result, this research will use the Time-Varying DID Model to examine the efficiency of EP policy in fostering regional sustainable development while also distinguishing the starting times of various pilot provinces. The starting time specified in each province’s *Outline of the EP Construction Planning* shall be used when determining the timing of the implementation of pilot provincial policies. The model will use a two-way fixed Time-Varying DID Model to eliminate both individual and temporal differences. The fundamental model is characterized by:(1)$$COD_{it}=\alpha_{0}+DID*Treat_{i}*Post_{it}+\alpha *Z_{it}+\mu_{i}+\tau_{t}+\varepsilon_{it}$$
(2)$$SO_{2it}=\beta_{0}+DID*Treat_{i}*Post_{it}+\beta *Z_{it}+\mu_{i}+\tau_{t}+\varepsilon_{it}$$
where $COD_{it}$ is the province *i*’s emission intensity of COD for the year *t*. $\;SO_{2it}$ is the province *i*’s emission intensity of SO_2_ for the year *t*. The terms $\;\alpha_{0}$ and $\beta_{0}$ are constant. The policy dummy variable called $Treat_{i}$ has a value of 1 if the province *i is* a pilot EP and 0 otherwise. The time dummy variable $Post_{it}$ indicates whether the policy is implemented in the province *i*’s year *t*. If the policy is implemented, it equals 1 and 0 otherwise. The net result of implementing the policy is DID, which is the coefficients of $Treat_{i}*Post_{it}$. A negative and significant DID indicates that the policy decreases the pollutant emission intensity, whereas a positive and significant DID means that the policy increases the pollutant emission intensity. When DID is not significant, it suggests that EP policy did not have an impact on the COD emission intensity and SO_2_ emission intensity. A number of control factors are represented by $Z_{it}$ that could influence the pollutant emission intensity. $\mu_{i}$ is the province fixed effect, $\tau_{t}$ is the time fixed effect, $\varepsilon_{it}$ represents the standard residual term.

### 2.2. Mediating Effect Analysis 

If the outcome is significant, this paper will examine the mechanism of this policy effect in more detail. In the world of economics, mediating effect analysis has become very popular recently. It can aid in the analysis of the mechanism and process through which policy works. We may obtain the more insightful research findings of the policy mechanism if the efficient mediating variable effect relationship can be identified.

When a particular variable takes into account the relationship between the predictor and the dependent variable and discusses how or why such effects happen, it is considered to serve as a mediator. The classical causal steps approach popularized by Baron and Kenny [58], is frequently employed in academic settings to investigate mediating impact. However, it can only be used to test the situation of a single mediator and cannot be used to test the case of several mediators [59,60]. Accordingly, this work employed the Bootstrapping method to examine the specific and overall indirect effect in the case of multiple parallel mediation variables, in accordance with the research of Preacher and Hayes [60]. Bootstrapping is a computationally intensive method that involves repeatedly sampling from the data set and estimating the indirect effect in each resampled data set. This strategy has several appealing qualities: First, we may compare the sizes of the distinct indirect effects. Secondly, we can assess the overall indirect effects of multiple mediating variables simultaneously. Third, we can determine if there are any missing mediation paths.

The multiple mediator model is represented by following regression equations:(3)$$COD_{it}\left(SO_{2it}\right)=\gamma +cTreat_{i}*Post_{it}+e_{1}$$
(4)$$M_{it}=\gamma +aTreat_{i}*Post_{it}+e_{2}$$
(5)$$COD_{it}\left(SO_{2it}\right)=\gamma +c^{\prime }Treat*Post_{it}+bM_{it}+e_{3}$$
where $COD_{it}(SO_{2it})$ represents the pollutant emissions intensity of COD or SO_2_, and $Treat_{i}*Post_{it}$ is the previously indicated policy variable. *c* represents the overall indirect effect of policy on the dependent variables COD and SO_2_ emission intensity. $M_{it}$ is a collection of mediation variables that indicate composition effect, technique effect, scale effect (the industry output value proportion, GDP per capita, number of invention patents per 10^4^ people, urbanization rate, and population density of the province *i* in year *t*), as well as investment in industrial pollution control, local financial expenditure on environmental production, and pollution discharge fees. *a* is the influence of the independent variable of policy implementation on the mediator variable $M_{it}$. *b* indicates the effect of mediators on dependent variable after eliminating the influence of independent variable on dependent variable. $c^{\prime }$ denotes the direct effect of the independent variable of policy implementation on the dependent variable after controlling the influence of the mediating variable. γ is the constant term, *e*_1_, *e*_2_, and *e*_3_ are the random error terms. The other variables are identical to those in the text’s earlier section.

To ascertain whether an intermediary path exists, we must examine the statistical significance of a·b. The study hypothesis’s suggested mediation path exists if a·b is significant. If not, the path of mediation is nonexistent. Additionally, testing the entire effect *c* is not required when the Bootstrapping approach is employed to assess the mediation effect.

### 2.3. Variables 

Explained variables: The intensity of SO_2_ and COD pollution emissions in each province are the explained variables in this paper. Pollutant emission intensity, which measures the environmental load per unit of newly created economic value, is the pollutant emission per unit of GDP. The ratio of COD and SO_2_ emissions to the actual total output value of each province are taken as the explanatory variables using their respective logarithms. To determine the comparable real GDP statistics for each year, the GDP data in this study used constant price adjustment in the year 2000.

Explanatory variables: The analysis of the Time-Varying DID Model in this research is based on provincial panel data. The primary explanatory factors in this study are the terms that interact between the policy dummy variable and the time dummy variable. It is the overall result of the application of policy. In order to examine the impact of policy implementation, this study will concentrate on the direction and significance of the interaction terms.

Control variables: The endogenous and fixed effects of region and time were effectively controlled by the DID of panel data. We also choose a number of control variables in order to further limit the deviation brought on by omitted variables and manage the impact of other factors on the explained variables. Environmental contamination is primarily impacted by the composition effect, technique effect, and scale effect of human economic activity [61]. The composition effect depicts how the environment is impacted by social and economic structure. The economic composition is determined by *the percentage of industrial production value in the overall social output value*, whereas the social composition is determined by the *urbanization rate*. The technique effect is determined by *the number of invention patents per 10^4^ people*. *Population density* and *per capita GDP* are used to quantify the scale effect. The COD emission intensity, SO_2_ emission intensity, per capita GDP, population density, and number of invention patents per 10^4^ people were all processed logarithmically to lessen sample dispersion and the potential for heteroscedasticity.

Mediating variables: In this analysis of the mediating effect, we will examine the composition effect, technique effect, and scale effect to determine the way how EP policy works. At the same time, we have included several important indicators that may affect the implementation of the policy. They are, *investment in industrial pollution control* (mainly invested by enterprises), *local financial expenditure on environmental production* (mainly invested by local finance), and *pollution discharge fees*. There are a total of 8 mediating variables introduced: proportion of industrial output value, per capita GDP, number of invention patents per 10^4^ people, urbanization rate, population density, enterprise investment, local finance, and pollution discharge fees. The results of the mediation effect analysis are obtained using Preacher and Hayes’s [60], bootstrapping method of multiple intermediate variables.

### 2.4. Data and Resource

This research assesses the effect of EPs on local sustainable development using panel data collected from 31 Chinese provinces between 1997 and 2017. The original information was gathered from the *China Statistical Yearbook*, *the China Environment Yearbook*, *China Statistical Yearbook on Environment*, as well as the provincial statistical yearbooks and statistical bulletins. The sample period for this study is set between 1997 and 2017 for the two following reasons: (1) The first pilot EP was approved for Hainan in 1999, and the Hainan Eco-province Construction Plan was published in 2000. As a result, the year 2000 can be seen as the start of the EP policy, while the year three before to the policy can be regarded as the start of the data. (2) The second national census of pollution sources in China was launched in 2017. The environmental data after 2017 have not been released. Considering the availability of data, we use the data before 2017 to ensure the reliability of the conclusion. In view of data integrity and availability, the scope of this study does not include Hong Kong, Macao, and Taiwan. The summary statistics of variables are displayed in Table 1.

## 3. Results and Discussion

### 3.1. Evaluation of Environmental Policy Effectiveness

#### 3.1.1. Basic Estimation Results

The estimation outcomes for Time-Varying DID models 1 and 2 are shown in Table 2. The basic model’s regression test results without control variables are shown in columns (1) and (3). The regression findings with control variables like Industry, GDP, Pop, Urban, and Invention are shown in columns (2) and (4). The average treatment effects of COD emission intensity without and with control variables are −0.048 (*p* < 0.01) and −0.044 (*p* < 0.01), respectively, according to columns (1) and (2). The average COD emission intensities of EP will fall by 5.495 percent (0.0482/0.8874) and 4.99 percent (0.0438/0.8874) in comparison to the non-pilot provinces. The findings in columns (3) and (4) show that both without and with the addition of control variables, the structures of EP have a negative impact on SO_2_ emission intensities. The coefficients are −0.081 (*p* < 0.001) and −0.064 (*p* < 0.001), respectively. In comparison to non-pilot provinces, this suggests that the SO_2_ emission intensities of EPs can be lowered by 8.58 percent (0.0813/0.9478) and 6.77 percent (0.0642/0.9478). Regardless of whether control variables are included, all regression findings demonstrate that the pilot policies of EPs have a significant decrease effect on pollutant emission intensities. The link between the dummy factors for the policy effect and the intensity of the pollutant emission is still negative and significant with the addition of the control variables, although the suppression effects are marginally lessened.

Because ecosystem management is intricate and challenging to comprehend and administer, it is difficult for central institutions like national governments to plan for and exert control over it [62]. Many different players and intricate organizations are involved in environmental governance. Even in China, a nation with a strong central government, local governments may not be able to implement national policies effectively at the local level due to a lack of understanding on their part. It is typical for judicial powers to overlap. Therefore, in a society where social ecosystems are interconnected, it is crucial to be worried about the centralization or decentralization of environmental control [63]. Take the EP as an example. From planning to project execution to outcome evaluation, it involves a large number of participants, including businesses and individuals from a variety of industries, complicated multi-level organizations, and departments working in simultaneously. It is a system project that spans multiple regions, industries, and sectors. The degree and effectiveness of local government’s environmental control will be impacted by a variety of factors as regional economic development faces stronger competition. Therefore, there are administrative level and functional authority restrictions when depending solely on the environmental protection department of a prefecture level or smaller unit to promote unilaterally.

The leaders of the province governments and other relevant departments shall organize a leading group for each EP in accordance with the construction plans. This leading group is an effort to coordinate and address significant issues that arise during EP construction, particularly issues that cut across sectors, and to provide a system for accountability. It serves as a connecting entity that links the ecological management departments with other industries including the economics and energy. In order to accomplish the sustainable development objectives of provincial areas, the provincial government can now coordinate and constrain the building of EP to other parallel departments instead of being a departmental responsibility of the environmental protection department. This variance in organizational structure may be one of the important factors why provincial policies effectively reduce pollutant emission intensity.

#### 3.1.2. Model Robustness Test

##### Parallel Trend Test and Dynamic Effect Analysis

Assuming that the parallel trend assumption is met, the traditional DID method is applicable. Prior to the adoption of the policy, there shouldn’t be any discernible differences in the trends of pollutant emission intensities between the treatment group and the control group. The parallel trend assumption must also be met by the Time-Varying DID approach. This paper uses the method of event analysis to test the parallel trend based on the practice in Alder, Shao [64]. For parallel trend analysis, we use the 11 years of data, from 3 years prior to the policy to 7 years following the policy (Figure 3). It displays the estimated values of the regression coefficients for the significant levels of both COD emission intensity and SO_2_ emission intensity. Before applying EP policies, the predicted coefficient values fluctuate about zero, and the confidence intervals also encompass zero. This shows that we are unable to reject the null hypothesis of zero. Pollutant emission intensities between the treatment group and the control group prior to EP policy are not significantly different. The parallel trend hypothesis is supported by the research sample used in this paper.

The dynamic impact of the policy implication is also shown in Figure 3. From the starting year (t) to 4 years after the adoption of the policy, the coefficients show no discernible variation. The policy effect coefficient of COD and SO_2_ emission intensity, however, declined to a significant negative value starting five years after the program’s adoption. This demonstrates that the impact of EP policy has a five-year lag.

##### Placebo Test 

Since the intensity of pollutant emissions can vary depending on many variables. Other significant national political, economic, and environmental factors not included in this work may have an impact on the effect of pilot EPs in basic regression on pollutant emission intensity. This paper uses the placebo test method of replacement samples in order to prevent the statistically significant findings from the conclusion from the aforementioned random causes. To create the falsification test, we chose non-pilot provinces as the samples for the placebo treatment group. If the same outcome is not observed in the provinces and areas of the placebo sample, it suggests that the EP’s pilot policies, rather than other laws and regulations in the same time frame, are what are responsible for the drop in pollutant emission intensity in the original treatment group. In this study, the surrounding seven non-pilot provinces (Anhui, Fujian, Hebei, Guangxi, Sichuan, Shanxi, and Hubei) were used in place of the original treatment group’s seven pilot provinces (Jiangxi, Guangdong, Inner Mongolia, Yunnan, Qinghai, Shaanxi and Hunan). Regression is performed using models (1) and (2) to produce the outcomes shown in Table 3. The estimated value of DID of the policy effect coefficients is not significant, according to the regression results in columns (1) to (4) of the placebo test. When compared to the treatment group in the actual pilot Eps, the provinces in the placebo sample do not exhibit the same significant adverse effects of pollutant emission intensities. The fact that the aforementioned falsification test can rule out the impact of other policies throughout the study period on pollutant emission reductions further demonstrates the validity of this paper’s result.

##### Year-by-Year PSM-DID Model Test 

There may be endogenous problems in the effect of EP pilot on regional pollutant emission. The provinces chosen to be EP pilots were not chosen at random, and they might differ from non-pilot provinces in some ways. The effect estimation of the policy will be inaccurate if the non-randomness of EP’s pilot policy is not taken into account. Propensity score matching (PSM) and difference-in-differences (DID) are therefore coupled to estimate policy effects by drawing on other literature approaches [65]. In order to execute the Time-varying DID test, a specific strategy is to match the non-pilot provinces with the pilot provinces year after year using similar features. Control variables are used as covariates in the PSM matching process of model testing, and the balance of data matching is then tested. The absolute values of variances are all less than 10%, and the T values are not significant, according to the experimental data in Table 4. After score matching, the data balancing test of the EP pilot passed, and it is established that the variables chosen for the propensity values match are adequate. The independent distribution requirement is met, and the propensity value matching result is appropriate.

According to the PSM-DID approach, Table 5 shows the year-by-year pollutant emission effects of the EP pilot policy. When no control variable is included, the regression findings of columns (1) and (3) show that the policy has net effects of −0.091 (*p* < 0.01) and −0.119 (*p* < 0.001). After adding control variables, the results of columns (2) and (4) show that the net effects of the EP pilot policies are −0.071 (*p* < 0.05) and −0.097 (*p* < 0.01). This demonstrates that the EP policy considerably reduces the intensity of pollutant emissions using the PSM-DID technique year by year, and the effect is larger than that under the Time-Varying DID method, further supporting the empirical conclusion of this research.

### 3.2. Heterogeneity Analysis

Due to the clear geographical differences in economic development in China, eastern China is more developed than central and western China. Starting in the eastern provinces, the EP pilot project was subsequently expanded to the central and western regions. In this essay, the various effects of EPs on these three locations will be further examined. Table 6 displays the outcomes based on the model developed above.

Table 6 demonstrates that the net effect DID of EP policy exhibits substantial heterogeneity in the three locations following the incorporation of control variables. After the policy’s implementation, the western and central provinces see a significant reduction in COD and SO_2_ emission intensities, but not the eastern provinces. From the regression coefficient of the control variables, the eastern region’s COD emission intensity is significantly correlated with the share of industrial output value, per capita GDP, and population density, while the share of industrial output value, per capita GDP, and the number of patent applications per 10^4^ people are significantly correlated with the share of SO_2_ emission intensity. However, there is no statistically significant impact of EP policy in eastern China.

By observing the pollutant emission intensities of all provinces and three regional pilot provinces in China from 1997 to 2017 (Figure 4), the average index of the eastern pilot provinces is always lower than the average of all provinces in China, while the average index of the western and central pilot provinces is first much higher than the national average, and then gradually drops to near the national average. This reveals that the EP policy has a significant inhibitory effect on regions with high pollutant emission intensity, but not on regions with low pollutant emission intensity. Although the policy has an overall inhibitory effect on pollutant emission intensities, it has substantial geographical variation.

In the EP construction indicators, regionalized differences are made for some economic indicators (annual net income per capita in rural and urban areas). But for the indicators of ecological environmental protection and social progress, a national unified indicator system is adopted. As can be seen from Figure 4, for the developed eastern regions, the intensity of pollutant emissions is lower than the national average. Therefore, for these regions, there is not enough incentive to innovate or reduce the intensity of pollutant emissions. This may be one reason of why the impact of EP policy on pollutant emission intensity is difficult to work in the eastern regions. In contrast, the developed eastern regions have a higher share of total pollutant emissions in the country due to their dense industries and high population density. Therefore, in the policy formulation, setting differentiated construction targets for different regions will help mobilize the motivation of each region in the country to reduce pollutant emission intensity.

Although provincial government agencies are responsible for the execution of EPs, policy-making departments should develop pertinent policies to carry out the objectives of EPs in the context of their particular regional circumstances [66]. For regions with relatively developed market economies, in addition to rule of law and administrative means, greater use of market-based means is needed to promote environmental governance in order to achieve the goals of EPs.

### 3.3. Mechanism Analysis and Discussion

The mediating impact of EP policy is seen in Table 7 and Table 8. The findings demonstrate that:

(1) Total mediating effect (Table 7): Zero is not contained in the total effect intervals (BootLLCI, BootULCI), and the total indirect effects are significantly negative. The total effect interval of COD emission intensity is (−0.1751, −0.058) and the effect size is −0.1171. The total effect confidence interval of SO_2_ emission intensity is (−0.2252, −0.0276), the effect size is −0.1254. This finding demonstrates that the policy, in combination with this set of intermediary variables, greatly reduces the intensity of pollutant emission.

(2) Independent mediating effect (Table 7): In terms of COD emission intensity, GDP, and Invention have significant negative mediation effects that will lower COD emission intensity. The percentage of industrial production, Investment in industrial pollution control will significantly raise the intensity of COD emission. Population density, Urbanization rate, Pollution discharge fees, and Local financial expenditure on environmental protection have no mediating effect of the EP policy. GDP, invention, and population density all have significant negative mediating impacts on SO_2_ emission intensity that will reduce it. The intensity of SO_2_ emissions is actually increased by the Rate of Urbanization, Pollution discharge fees, and Investment. Local financial expenditure has no mediation impact in this situation. (3) Direct effect of the policy (Table 8): If the policy effect is no longer significant after the mediator variables are controlled, it means that all of the mediator variables assumed in the model are present and that no additional mediator variables are missing. If the policy effect is still significant, it suggests that additional mediating factors that were left out may still be present [59]. Therefore, the policy effect on COD emission intensity is no longer significant, but the situation with SO_2_ is different. That indicates the mediators we chose have the complete mediating effect of the policy on COD emission intensity, and partial mediating effect on SO_2_ emission intensity.

The EP policy has a significant inhibitory effect on pollutant emission intensity. GDP and Invention are the most important mediators of the reduction of pollutant emission intensity by the EP policy. This reflects the pushing efforts of entrepreneurial innovation and production [12].The results of the mediating effect model show that there is much room for policy optimization. 

(1) The EP policy has raised the proportion of industrial industries in the pilot provinces, which has slightly raised the intensity of COD and SO_2_ emissions. China’s industrialization has progressed quite unevenly across the country. The central and western regions are only in the early and intermediate stages of industrialization, whereas the eastern region has reached the post-industrialization stage. The EP policy’s implementation has sped up industrialization in the central and western areas and increased the proportion of industrial industries, leading to a minor increase in the intensity of pollutant emissions.

(2) Local financial expenditure on environmental protection have little media effect of the EP policy. Although each pilot EP has established a Construction Plan, local financial resources for environmental protection are still insufficient. Fiscal spending for environmental preservation in China has been maintained at a low level, often only about 2.50% of total fiscal expenditure. Due to the lack of financial investment by local governments, some of the completed pollution-control facilities are out of operation or under-operating due to the failure to guarantee the operating costs [67]. The central government accounts for less than 3% of the total state expenditure on environmental protection. The incentive provided by the fiscal funds is not enough to drive more social capital to join. 

(3) Industrial pollution control investment instead increases the intensity of pollutant emissions. The causes of this might be attributed to two factors. First, the pilot Eps local governments’ twin objectives of promoting economic development and environmental protection have resulted in a decrease in industrial businesses’ investment in environmental protection. According to recent studies, regional economic growth pressures in areas with high financial stress and low environmental stress have undermined industrial businesses’ investments in environmental protection more than other regions [68]. Second, pilot Eps’ industrial businesses lack the professionalism and investment efficiency to handle pollution on their own. The market for professional environmental protection enterprises is still immaturely developed and lacks a professional approach to pollution control. Because of this, even though industrial enterprises have made investments in environmental treatment, the effect is not immediately apparent but does increase costs for businesses, move the market’s supply–demand curve to the left, lower GDP, and intensify pollutant emissions.

(4) Pollution discharge fees of Eps have little mediating effect on COD emission intensity but increase the SO_2_ emission intensity. Pollution discharge fees are considered by economists to be one of the most effective policy tools [69]. China started to impose sewage charges on enterprises and individuals who emit pollutants in 2003. Although the collection of emission fees has played a role, they cover only 73% of industrial enterprises as of 2015 due to their administrative nature. The standard and implementation of emission fees also need to be improved. When the emission fee was implemented, there was a situation where “bad money drives out good money”, and the cost of energy saving and emission reduction for enterprises with low emissions was higher than the emission fee paid by enterprises with high emissions, which put these enterprises with high energy saving and emission reduction costs at a disadvantage in the market. In the pilot EPs, industrial enterprises are more inclined to choose to pay emission fees than to invest or innovate in environmental protection in order to obtain the lowest cost of pollutant emissions, which in turn increases the intensity of pollutant emissions [70]. Starting in 2018, the country stopped collecting pollution discharge fees and replaced them with an environmental protection tax, which floats according to different regions and the concentration of pollutants emitted. This move may reduce the drawbacks of the previous charges.

## 4. Conclusions 

China’s overall pollution emissions and pollution intensity decreased steadily from 1997 to 2017. In comparison to the non-pilot provinces, the pilot provinces’ COD and SO_2_ emission intensity has decreased dramatically since the EP policy’s adoption. After the EP policy is put into place, it aids China in easing the burden of economic growth on the environment. The impact of policies varies by region, though. It matters in the western and central pilot provinces since they initially have relatively high pollutant emission intensities. However, the policy effect is not significant in reducing pollutant emission intensities in the east pilot provinces whose pollutant emission intensities are lower than the national average.

By analyzing the mechanism of policy action, we find that the EP policy considerably reduces pollutant emission intensities mainly through increasing GDP per capita and invention, which is the main mediating pathway of policy action. At the same time, the policy also raised the pollutant emission intensity in the pilot provinces by increasing the proportion of industrial industries and reducing the investment in environmental protection by industrial enterprises. The collection of pollution discharge fees in the pilot provinces resulted in an increase in SO_2_ emission intensity but no discernible impact on COD emission intensity. In the pilot provinces, local financial investments in environmental protection had no significant effect on the index.

Based on the above findings, we put forward several policy suggestions: First, different construction targets should be set for different regions. More challenging goals and more market-oriented means should be utilized to boost the incentive to reduce emissions in economically developed locations with high pollution emissions. Second, local finances in the pilot provinces should be more inclined to invest in environmental protection. Enhance the long-term viability of local government environmental protection programs by improving their sustainable operation. Third, the pilot provinces should promote innovation in investment and financing mechanisms for the environmental protection industry. By increasing government investment in environmental protection and government procurement, improve the environmental protection industry venture capital mechanism to attract more social capital to enter the environmental protection industry and broaden the sources of funding and cooperation models for environmental protection investment.

There are still some missing mediating variables in SO_2_ emission intensity. In order to thoroughly understand the mechanism of the policy, more studies will be required in the future that will add more mediating variables to the model. Whether the implementation of EP can support other aspects of social and economic growth also requires further research. 

## Figures and Tables

**Figure 1 ijerph-19-11025-f001:**
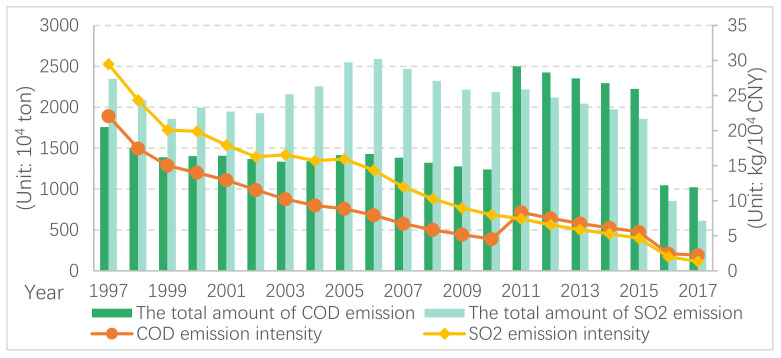
Pollutant emission intensities of China (1997–2017).

**Figure 2 ijerph-19-11025-f002:**
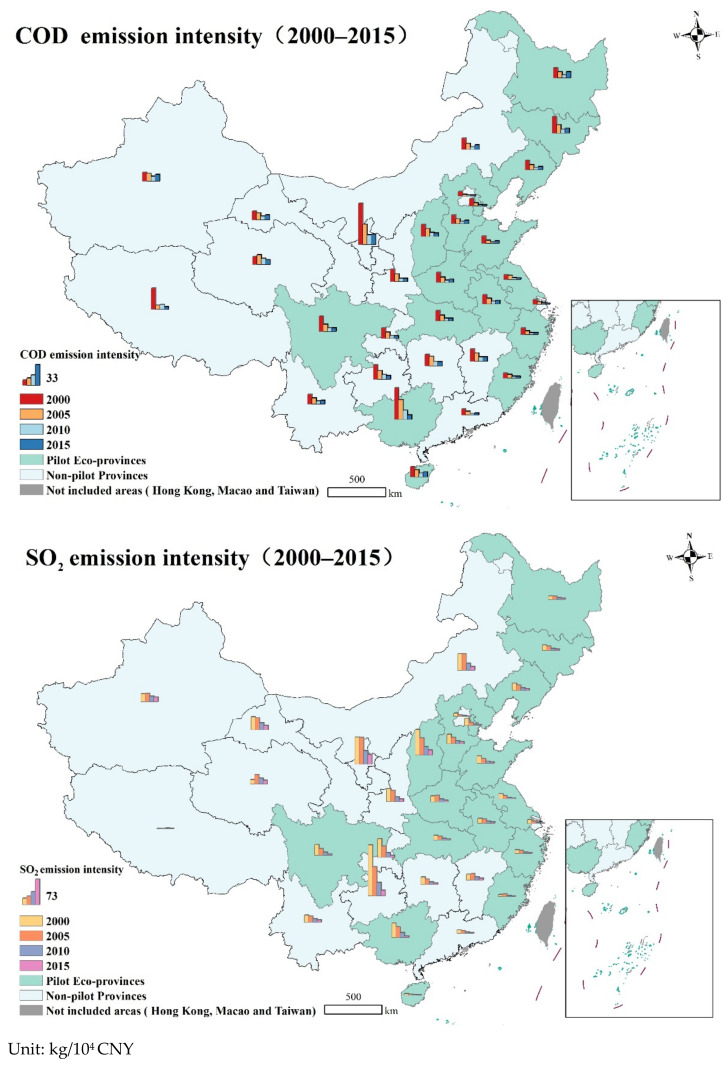
Pollutant emission intensities of provinces in China (2000–2015).

**Figure 3 ijerph-19-11025-f003:**
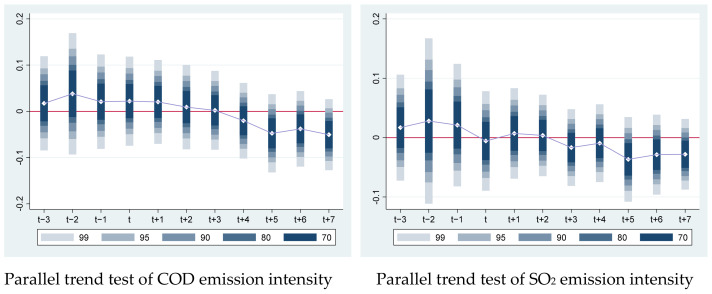
Dynamic effect of parallel trend test.

**Figure 4 ijerph-19-11025-f004:**
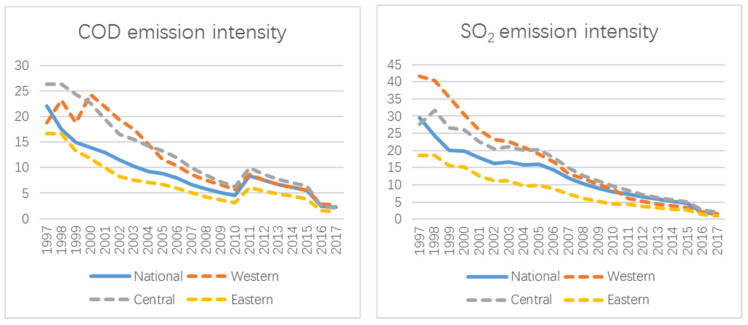
Pollutant emission intensities in different regions.

**Table 1 ijerph-19-11025-t001:** Main Variables, meanings, and descriptive statistics.

Variables	Meanings	Measures	Mean	S.D.	Min.	Max.	Obs.
COD	COD emission intensity	COD total emissions/GDP, (log)	0.8774	0.3463	−0.20	1.82	651
SO_2_	SO_2_ emission intensity	SO_2_ total emissions/regional GDP, (log)	0.9478	0.5302	−1.08	2.36	651
Treat	treatment variable	0 in control group, and 1 in treatment group	0.52	0.5001	0	1	651
Post	time variable	0 before the policy, 1 after the policy	0.32	0.4666	0	1	651
DID	policy effect	Treat × Post cross term coefficient	0.32	0.4666	0	1	651
Industry	Proportion of industrial output	Industrial GDP/regional GDP	0.42433	0.0853	0.106	0.620	651
GDP	per capital GDP	Regional GDP/population, (log)	4.1647	0.3388	3.34	4.96	651
Pop	population density	Population/area of region, (log)	3.4440	0.4760	0.30	4.05	651
Urban	urbanization rate	Urban population/resident population	0.47165	0.1641	0.140	0.896	651
Invention	Number of patents authorized per 10^4^ people	The number of patent applications/resident population, (log)	3.6250	0.8153	0.78	5.52	651
Investment	Investment in industrial pollution control	Investment in industrial pollution control, (log)	4.8294	0.6931	0.04	5.99	558
Finance	Local financial expenditure on environmental protection	Local financial expenditure on environmental protection, (log)	1.7291	0.3238	0.62	2.52	341
Charge	pollution discharge fees	pollution discharge fees, (log)	4.4015	0.6465	−0.14	5.36	434

**Table 2 ijerph-19-11025-t002:** Eco-province pilot pollutant emission intensity treatment effects—Time-Varying DID Model.

	(**1**)	(**2**)	(**3**)	(**4**)
	**COD**	**COD**	**SO_2_**	**SO_2_**
DID	−0.048210 **	−0.043779 **	−0.081282 ***	−0.064179 ***
	(−3.18)	(−3.24)	(−4.36)	(−4.08)
Industry		0.697823 ***		1.336705 ***
		(4.55)		(5.99)
GDP		−0.375814 **		−0.348993 *
		(−2.89)		(−2.03)
Invention		−0.084425 *		−0.221217 ***
		(−2.18)		(−6.25)
Urban		0.345314		0.009465
		(1.59)		(0.03)
Pop		0.032766		0.093437
		(0.54)		(1.89)
_cons	0.892807 ***	2.190760 ***	0.973760 ***	2.330195 **
	(157.18)	(3.95)	(123.65)	(3.18)
Individual effect	Control	Control	Control	Control
Time effect	Control	Control	Control	Control
N	651	651	651	651
adj. R-sq	0.9099	0.9161	0.9392	0.9501
AIC	−1.10 × 10^3^	−1.20 × 10^3^	−8.50 × 10^2^	−9.80 × 10^2^
BIC	−1.10 × 10^3^	−1.20 × 10^3^	−8.40 × 10^2^	−9.40 × 10^2^

Notes: T statistics in parentheses; * *p* < 0.05, ** *p* < 0.01, *** *p* < 0.001.

**Table 3 ijerph-19-11025-t003:** Placebo test.

	(**1**)	(**2**)	(**3**)	(**4**)
	**COD**	**COD**	**SO_2_**	**SO_2_**
DID	−0.020965	−0.010025	0.014192	0.030752
	(−1.39)	(−0.73)	−0.74	−1.8
Industry		0.718208 ***		1.417141 ***
		(4.69)		(6.23)
GDP		−0.386206 **		−0.448885 **
		(−2.98)		(−2.60)
Invention		−0.085913 *		−0.217291 ***
		(−2.21)		(−6.10)
Urban		0.292614		−0.082184
		(1.35)		(−0.29)
Pop		0.035914		0.093743 *
		(0.6)		(2.01)
_cons	0.884102 ***	2.234014 ***	0.943255 ***	2.709691 ***
	(153.76)	(4.03)	(113.12)	(3.73)
Individual effect	Control	Control	Control	Control
Time effect	Control	Control	Control	Control
N	651	651	651	651
adj.R-sq	0.9087	0.915	0.9376	0.9493
AIC	−1.10 × 10^3^	−1.20 × 10^3^	−8.30 × 10^2^	−9.70 × 10^2^
BIC	−1.10 × 10^3^	−1.20 × 10^3^	−8.30 × 10^2^	−9.30 × 10^2^

Notes: T statistics in parentheses; * *p* < 0.05, ** *p* < 0.01, *** *p* < 0.001.

**Table 4 ijerph-19-11025-t004:** PSM matching balance test.

	Mean			*T* value	
Variables	Treatment Group	Control Group	Bias (%)	t	*p* > |t|
Industry	0.4177	0.4122	7.2	0.63	0.530
GDP	4.1667	4.1718	−1.5	−0.12	0.907
Invention	3.6383	3.6464	−1.1	−0.09	0.928
Urban	0.4879	0.4820	3.5	0.28	0.783
Pop	3.4317	3.4634	−7.6	−0.64	0.525

**Table 5 ijerph-19-11025-t005:** Treatment effect: Year-by-year PSM-DID test.

	(**1**)	(**2**)	(**3**)	(**4**)
	**COD**	**COD**	**SO_2_**	**SO_2_**
DID	−0.090572 **	−0.071231 *	−0.119034 ***	−0.097199 **
	(−3.21)	(−2.44)	(−3.63)	(−3.07)
Industry		0.669147 **		0.839463 **
		(2.97)		(2.81)
GDP		−0.721733 **		−0.410014
		(−3.15)		(−1.73)
Invention		−0.012731		−0.157032 *
		(−0.17)		(−2.04)
Urban		0.455284		0.714038
		(1.23)		(1.34)
Pop		0.044086		0.133505
		(0.57)		(1.88)
_cons	0.926592 ***	3.321187 **	0.982756 ***	2.097971
	(87.36)	(3.3)	(72.11)	(1.71)
Individual effect	Control	Control	Control	Control
Time effect	Control	Control	Control	Control
N	221	221	221	221
adj. R-sq	0.9246	0.9298	0.9404	0.946
AIC	−4.30 × 10^2^	−4.50 × 10^2^	−3.40 × 10^2^	−3.60 × 10^2^
BIC	−4.30 × 10^2^	−4.20 × 10^2^	−3.30 × 10^2^	−3.30 × 10^2^

Notes: T statistics in parentheses; * *p* < 0.05, ** *p* < 0.01, *** *p* < 0.001.

**Table 6 ijerph-19-11025-t006:** Comparison of EP policy effects in different regions.

	COD				SO_2_	
	Western	Central	Eastern	Western	Central	Eastern
DID	−0.1111 ***	−0.0706 ***	−0.0313	−0.2191 ***	−0.0926 ***	−0.0325
	(−3.47)	(−3.61)	(−1.86)	(−4.93)	(−3.91)	(−1.76)
Industry	0.088103	0.6657 ***	1.0900 ***	−0.0539	1.52902 ***	2.3525 ***
	(1.17)	(3.65)	(5.69)	(−0.49)	(6.08)	(9.45)
GDP	−0.7934 ***	−0.8827 ***	−0.8647 ***	−0.1206	−0.7842 ***	−0.7807 ***
	(−3.33)	(−4.22)	(−5.18)	(−0.37)	(−3.82)	(−4.50)
Invention	−0.1000	−0.0464	−0.0616	−0.2406 **	−0.2317 ***	−0.1498 ***
	(−1.53)	(−0.91)	(−1.32)	(−2.81)	(−4.98)	(−3.70)
Urban	0.0571	0.0909	0.3174	0.0559	−0.640192	0.16125
	(1.76)	(0.3)	(1.2)	(1.12)	(−1.71)	(0.55)
Pop	−2.0 ***	−1.3 ***	−0.9876 ***	−1.1	−1.4 **	−0.4193
	(−4.95)	(−3.85)	(−3.76)	(−1.91)	(−2.97)	(−1.26)
_cons	8.4709 ***	7.2796 ***	6.3145 ***	4.6642	7.6953 ***	4.6291 ***
	(4.91)	(4.82)	(5.42)	(1.94)	(4.37)	(3.43)
Individual effect	Control	Control	Control	Control	Control	Control
Time effect	Control	Control	Control	Control	Control	Control
N	336	462	483	336	462	483
adj. R-sq	0.8978	0.9017	0.9226	0.9269	0.9445	0.9565
AIC	−5.50 × 10^2^	−7.80 × 10^2^	−9.20 × 10^2^	−3.10 × 10^2^	−6.10 × 10^2^	−7.30 × 10^2^
BIC	−5.30 × 10^2^	−7.50 × 10^2^	−8.90 × 10^2^	−2.80 × 10^2^	−5.80 × 10^2^	−7.00 × 10^2^

Notes: T statistics in parentheses; ** *p* < 0.01, *** *p* < 0.001.

**Table 7 ijerph-19-11025-t007:** The mediating effect of Eco-province policy.

	COD Emission Intensity				SO_2_ Emission Intensity			
	Effect	BootSE	BootLLCI	BootULCI	Effect	BootSE	BootLLCI	BootULCI
TOTAL	−0.1171	0.0305	−0.1751	−0.058	−0.1254	0.0506	−0.2252	−0.0276
Industry	0.0232	0.0094	0.0082	0.045	0.0603	0.0181	0.0271	0.0979
GDP	−0.0899	0.0251	−0.1429	−0.0446	−0.2121	0.0495	−0.3174	−0.1249
Invention	−0.0581	0.0232	−0.1063	−0.0154	−0.1207	0.0324	−0.1887	−0.0639
Urban	−0.0035	0.0075	−0.0201	0.0108	0.0355	0.0209	0.0041	0.0838
Pop	−0.0234	0.0208	−0.0668	0.0149	−0.0766	0.0256	−0.1309	−0.0296
Finance	0.0022	0.0047	−0.0063	0.0128	0.0112	0.0091	−0.0011	0.0345
Charge	0.0088	0.013	−0.022	0.0297	0.0812	0.0163	0.0538	0.1174
Investment	0.0237	0.0107	0.0018	0.0445	0.0957	0.0205	0.0568	0.1372

**Table 8 ijerph-19-11025-t008:** Direct effect of Eco-province policy (after control mediating variables).

	Effect	se	t	*p*	LLCI	ULCI
COD	0.0366	0.0255	1.4338	0.1526	−0.0136	0.0868
SO_2_	−0.0835	0.0277	−3.0153	0.0028	−0.138	−0.029

Notes: The above data were calculated and listed though SPSS software by Bootstrap method. Among them, the sample size was 5000, and the 95% confidence interval was set.

## Data Availability

The data supporting the results presented in this study can be obtained from the corresponding author on request.

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
