# Peer review of "Does China’s Eco-Province Policy Effectively Reduce the Pollutant Emission Intensities?"

_ijerph, 2022, doi:10.3390/ijerph191711025_

Round 1

Reviewer 1 Report

The paper applied a Time-Varying DID Model by employing Chinese provincial panel data to quantitatively study the policy effect, and further utilized Mediating Effect Model to analyze the mechanism. In general, appropriate research ideas and methods, and found some useful conclusions. 

My suggestion is that some strategies for EP development and implementation should be proposed based on the findings of the research. 

There are some technical errors in the author's citation. Such as Page 3, line 112. The full text needs to be carefully checked.

Reviewer 2 Report

Dear authors,

I enjoyed reading your paper. The structure and data analysis of the paper is clear. You can view my detailed review from the attachment.

Round 2

Reviewer 2 Report

Dear authors,

I enjoyed reading the second version. I think it is ready to be published.

Best,

The reviewer